# Quality 4.0 in Action: Smart Hybrid Fault Diagnosis System in Plaster Production

**Javaneh Ramezani * and Javad Jassbi**

Faculty of Sciences and Technology and Uninova CTS, NOVA University of Lisbon, Campus de Caparica, 2829-516 Caparica, Portugal; j.jassbi@uninova.pt

* Correspondence: m.ramezani@campus.fct.unl.pt or ramezanijavaneh@gmail.com

**Abstract:** Industry 4.0 (I4.0) represents the Fourth Industrial Revolution in manufacturing, expressing the digital transformation of industrial companies employing emerging technologies. Factories of the future will enjoy hybrid solutions, while quality is the heart of all manufacturing systems regardless of the type of production and products. Quality 4.0 is a branch of I4.0 with the aim of boosting quality by employing smart solutions and intelligent algorithms. There are many conceptual frameworks and models, while the main challenge is to have the experience of Quality 4.0 in action at the workshop level. In this paper, a hybrid model based on a neural network (NN) and expert system (ES) is proposed for dealing with control chart patterns (CCPs). The idea is to have, instead of a passive descriptive model, a smart predictive model to recommend corrective actions. A construction plaster-producing company was used to present and evaluate the advantages of this novel approach, while the result shows the competency and eligibility of Quality 4.0 in action.

**Keywords:** statistical process control; control chart pattern; disruptions; disruption management; fault diagnosis; Industry 4.0; construction industry; plaster production; neural networks; decision support systems; expert systems; failure mode and effects analysis (FMEA); discriminant analysis

## 1. Introduction, Background, and Problem Statement

### 1.1. Introduction

In today's globally complex and competitive business environments, quality is one of the crucial issues for ensuring the success of enterprises [1]. In order to produce with the desired quality and meet the customer's expectations, production processes need to be monitored to avoid any defect and deviation [2]. Traditionally, statistical process control (SPC) was used as a powerful approach for monitoring and identifying variations manually [1,2]. Developments in manufacturing and information technology enabled SPC to move from merely statistical control to real-time diagnosis purposes with minimum human intervention [3]. Control charts, invented by Shewhart in the 1920s, are essential tools in SPC to assist in controlling the behavior of the process. These tools are used to decide if the process is behaving as intended or in the presence of some unnatural causes of variations. X-bar and R charts are basic Shewhart control charts for drawing a series of process measured data with control limits [3,4]. Process variation emerges from either common causes (natural variations) or specific causes (assignable reasons). Specific causes are those that cause changes and short-term fluctuations, and, if they occur, they destroy the stability of the process, which ought to be known and eliminated as quickly as time permits. Common causes are because of the inherent characteristics of the process, and, if they exist, deviations (background noise) are in control [5,6]. However, the most crucial ability of control charts is detecting various types of patterns consisting of a series of consecutive points that are observed on these charts, which reflects fluctuations in the process [7]. The control chart

patterns (CCPs) are generally divided into natural and unnatural patterns. Natural patterns usually exist in the manufacturing process and indicate that the process is statistically stable. As long as the measured data are inside the control limits or only natural random patterns exist, the process is under control. When some measurements fall out of the control limits or the measured data within the control limits signify a non-random pattern, the process is deemed out of control. Unnatural patterns displayed in control charts can be of various types, and each class can be related to specific causes unfavorably influencing the process stability. For example, "*Shift*" patterns may be related to variations in raw material, supplier, or machine, whereas "*Trend*" patterns may occur due to gauge wear or environmental changes [1,8]. Different common patterns that regularly emerge in control charts can be found in Figure 1. Over time, various further decision rules such as "*zone tests*" and "*run rules*", including "*Western Electric*," "*Nelson*", etc., were developed to assist quality control engineers and operators in detecting unnatural CCPs and circumstances leading to a change in the process [9]. Table 1 shows the most recommended rules for the Shewhart control charts to identify abnormal patterns and interpret their characteristic signs in the control chart. In general, the use of run rules can result in quickly signaling a shift in the process. However, the application of all these rules, when no particular cause exists, increases the risk of false alarms (Type I errors) to an unacceptable extent. In addition, run rules do not provide valuable pattern-related information because of a lack of sufficient pattern discrimination capability. Furthermore, control charts do not consider prior knowledge or adequate historical data. Therefore, these decision rules are not particularly useful for CCP recognition [10,11]. Since the analysis of control charts is complicated, because it relies on considerable statistical knowledge, skill, and experience of the practitioners (quality control personnel), developing an efficient automated pattern recognition system that can ensure steady and unbiased analysis of CCPs can compensate for this gap [12].

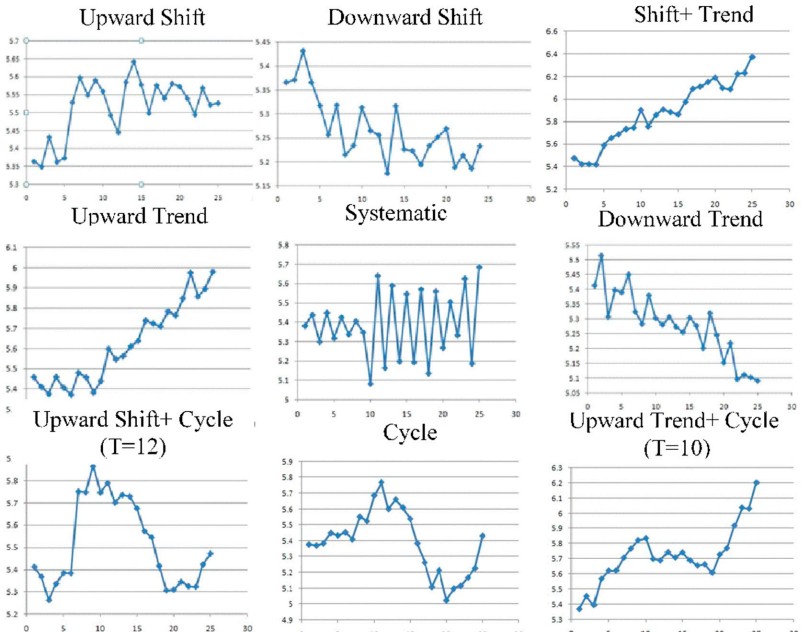

**Figure 1.** Typical patterns in control charts.

**Table 1.** Mostly recommended rules for detecting typical unnatural patterns.

| No. | Unnatural Pattern | Characteristic Signs in Control Chart |
|---|---|---|
| 1 | Over-control | Single point beyond control limits (above +3σ or below −3σ). |
| 2 | Shift | Sudden change (series of 9 points) above or below the central line. |
| 3 | Trend | Continuous movement (rise or fall) of 6 consecutive points. |
| 4 | Systematic | A point-to-point fluctuation (14 consecutive points alternating up and down). |
| 5 | Cycle | Periodic peaks and troughs (4 out of 5 points above +2σ or below −2σ). |
| 6 | Mixtures | A run of consecutive points on both sides of the central line, all far from the central line (8 points in a row more than +1σ from centerline). |

## 1.2. Background and Problem Statement

With the development in manufacturing and computing technology, several approaches were proposed using artificial intelligence technologies such as artificial neural networks (ANNs), expert systems (ESs), and fuzzy sets to automatically and intelligently CCP recognition [13]. In the domain of SPC, fast and accurate control, as well as observing the variation of quality characteristics and, consequently, recognition of unnatural patterns, is the primary purpose of each fault detection and diagnosis system. There are numerous studies in this field on CCP recognition that used different machine learning algorithms and other intelligent approaches, namely, K-nearest neighbors (KNN), decision trees (DT), NN-based models, ES-based models, support vector machine (SVM), wavelet-based models, and fuzzy logic [14–16]. These approaches aim at extracting meaningful information from a large amount of data to detect instabilities in the process with minimal time and cost and maximum accuracy [17]. To sum up, the most significant approaches are explained briefly in Table 2 by highlighting their advantages and disadvantages.

**Table 2.** Related works.

| Model | Advantage | Disadvantage | Related Works |
|---|---|---|---|
| KNN | - Very fast training (instance-based learning).<br>- Very easy to implement. | - Weakness in working with large dataset.<br>- Memory limitation.<br>- Sensitive to noisy data. | [15,18] |
| DT | - Simple to understand, interpret, and generate rules. | - May suffer from overfitting.<br>- Unstable classifier. | [15,16] |
| NN | - Does not need precise knowledge of interactions between the parameters.<br>- Learns to recognize patterns during the training phase.<br>- Able to handle noisy data.<br>- High performance. | - NN's topology cannot be systematically determined.<br>- Training of the network is prolonged, and processing for large NNs is difficult.<br>- Needs a large amount of useful training samples.<br>- Problem of overfitting. | [3,4,7,10–15,19–21] |
| ES | - Availability, consistency, extensibility, and testability of the information.<br>- Rules can be updated easily. | - Problems of incorrect recognition for similar statistical properties (features overlapping). | [9,12,14,19,22,23] |
| SVM | - Easily handles nonlinear, un/semi-structured, and high-dimensional data.<br>- Overfitting problem is not as much as other methods.<br>- With an appropriate kernel function, complex problems can be solved. | - Computationally expensive.<br>- Difficult understanding and interpreting the final model.<br>- Long training time for large datasets. | [2,14,24–26] |
| Fuzzy | - High precision. | - Low speed and the long run time of the system. | [5,14,21,26] |

The literature review shows that ANNs and ESs are the most widely used approaches, being easier to understand and implement and having higher performance in comparison to other CCP recognition approaches mentioned above. NNs are suitable for SPC as they are good at classification and pattern recognition, and they are able to handle the noisy measurements with no requirement for the provision of explicit rules regarding the monitored data [20]. Notably, ESs are useful for quality control applications due to their potential for identifying causes of deviations and recommending preventive and corrective actions [23]. There are two approaches to applying ANNs to CCP recognition: (1) using neural networks (NNs) to detect variation in X-bar and/or R charts, and (2) using NNs to identify unnatural patterns [19]. In this regard, NNs can be classified into two main categories: supervised NNs, involving multilayer perceptron (MLP) and radial basis function (RBF), and unsupervised NNs, including learning vector quantization (LVQ) and adaptive resonance theory (ART) [10]. Among the ANNs, the multilayer perceptron (MLP) was successfully exploited by many researchers in order to address the unnatural CCP recognition problem. Learning vector quantization (LVQ) is a well-applied alternative method to solve the problem of slowness in training the MLP network [4,12,20,27].

The fault diagnosis is an essential issue in SPC, to reduce downtime and disruption cascades that can ensue [24]. In recent years, various diagnostic systems were developed to automate fault diagnosis, but none of them fit our problem in the plaster production process discussed here. Most fault diagnosis approaches in the literature only considered a particular control chart, often X-bar or R (range) chart, to examine the process changes (mean or variance). However, in practice, in many processes, it is required to combine the two charts as multiple assignable causes may occur [28]. On the other hand, identification of unnatural patterns combined with specific knowledge of the process results in a more targeted diagnosis. Unfortunately, none of the CCP recognition models in the literature provide this combination automatically, which can be valuable for diagnostic purposes. Moreover, the performance of the model was not evaluated when developing these approaches in a real case study.

Yet, the common problem reported in these studies is the inability to recognize various single and concurrent CCPs, as well as a high rate of false recognition [4,29]. On the other hand, most applications of NNs and ESs to CCP recognition do not obtain more detailed information about the patterns and their change point (when these patterns are observed on control charts). This information is essential for practical assignable cause analysis and, in turn, accelerates the accomplishment of proper remedial activities [21].

Therefore, in this paper, designing a hybrid fault diagnosis system is proposed using NNs and a rule-based ES to help the quality control personnel in recognizing the roots of deviations, and in taking needed predictive or corrective actions. In the design process, for the structure of the NN, a modular approach comprising an LVQ network and seven multilayer perceptrons (MLPs) is used. Therefore, our work provides a neural expert system in intelligent real-time monitoring and predictive, corrective, and remedial diagnosis of process control in plaster production. To develop the proposed neural expert system (Figure 2), we address the following notable features of the model:

- Ability to detect various natural and unnatural (single and concurrent) CCPs.
- Monitoring and analyzing X-bar and R charts abnormalities simultaneously.
- Capability to estimate nonrandom patterns' corresponding parameters, different directions, and change points (starting point) in control charts.
- Identifying the responsible variables on the occurrence of unnatural patterns.
- Recognizing causes of process instability.
- Recommending predictive and/or corrective actions in a time of crisis.

The idea is to have, instead of a passive descriptive model, a smart predictive model to assist quality control engineers for the fault diagnosis of the process, particularly from a practical perspective regarding a Quality 4.0 era.

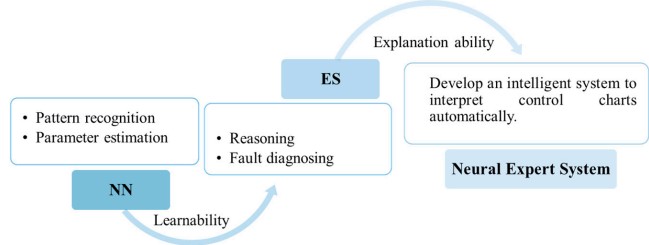

**Figure 2.** Combination of NN and ES: A neural expert system.

### 1.3. Contribution to Industry 4.0

Quality "4.0" is a branch of the Industry 4.0 (I4.0) movement associated with the digital transformation process connected with emerging technologies. Quality 4.0 could be defined as the application of Industry 4.0 technologies to quality management methods and tools [30]. According to Reference [31], "Quality 4.0 does not replace traditional quality methods, but rather builds and improves upon them". This concept covers all issues of advanced quality management in the digital era [32]. For quality (technology, processes, and people), Industry 4.0 enables the transformation of existing capabilities (culture, management, collaboration, and competencies) to drive value [31].

The impact of I4.0 on manufacturing is beyond just the physical production of goods, involving targeting all processes and functions to achieve flexibility, smartness, cost-effectiveness, and resilience. Artificial intelligence and machine learning are among the aforementioned technologies that can be utilized to enhance the quality as the heart of smart manufacturing [12,33,34].

On the other hand, construction projects face different sources of disruptions, as they are time-limited, expert-dependent, and highly influenced by process fluctuations caused by weather conditions, material quality, etc., which leads to a high level of complexity and uncertainty in the construction ecosystem [35]. Industry 4.0 challenged the construction industry ecosystem by demonstrating the construction digitalization potential for real-time data collecting, processing, and sharing tools to enhance alignment between demand and supply [35,36].

SPC is an essential tool to monitor process disruptions, safety assurance, and reliability analysis in construction projects [37]. Industry 4.0, with its automation, connectivity, and digital access capacity, is anticipated to be capable of increasing the efficiency and productivity of SPC. This could happen through enabling intelligent monitoring and diagnosis, automatic tracking of equipment and material, and real-time decision-making, especially in situations where the process is becoming more volatile and complex (Figure 3) [12,30].

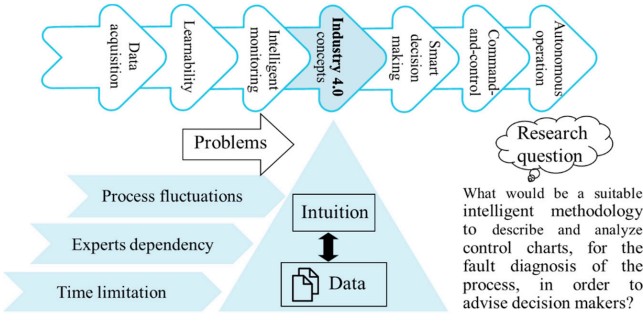

**Figure 3.** Quality 4.0: integration of traditional statistical process control (SPC) with Industry 4.0.

Digitalization and automation are the two pillars of smart manufacturing [38]. In this work, our effort was to develop a model based on traditional quality systems driven by disruptive technology. In this paper, artificial intelligence in the form of ANN and ES is employed. This proposes new values for the value chain of the manufacturing system on the factory floor level.

In fact, the innovation associated with digitalization, automation, communication, optimization, and customization of Industry 4.0 concepts and trends allows for real-time analysis and interpretation of production, industries, and service processes to improve quality by detecting failures and justifying possible causes while staying competitive in volatile business environments [30,39]. This work provides just a step to move forward and make the dream of "smart manufacturing" happen under the light of Industry 4.0.

The remainder of this paper is organized as follows: Section 2 concisely outlines the methodology of the research; Section 3 presents the proposed model; Section 4 describes the detailed structure of the model; Section 5 presents a comparative analysis and shows some results from a real case study; finally, Section 6 concludes the paper.

## 2. Materials and Methods

The methodology of this research is descriptive–experimental research, which is a systematic mapping study based on Reference [40] and an implemented case study in a plaster production plant. Figure 4 represents the schematic diagram depicting the proposed procedure. Using the review of the literature and benchmarking on the extraction of intelligent models used in process control, the structure of a hybrid fault diagnosis system using ANN and ES in the process control of plaster production is presented in this research. Mapping is used to present structuring to synthesize the three main research areas that include statistical process control, neural networks, and expert systems in this research. The case study, based on experiences from model implementation and validation in a plaster production plant, is reflected in Section 4. The plaster production process, which was selected as a case study, is a fluctuating process that has many influencing parameters. On the other hand, because the final production, i.e., construction plaster or so-called "plaster of Paris (PoP)", is mixed in a silo, the monitoring and modification processes in a short period can prevent the entire silo storage product from crashing. On the contrary, if the process is not monitored with statistical process control over an extended period, non-compliance of part of the product with the standard can crash the entire stored production. For example, in the case of filling more than 10% of the silo from a mismatched product, the whole product in the silo will crash. In order to improve the process quality, a survey was done of experts using a questionnaire and interviews to identify critical control parameters. The "initial setting time" of plaster was detected as the critical parameter of the production process. The initial setting time is dependent on the "crystal water" of baked plaster and ought to last between 7 and 15 min in the intended case study. The acceptance range in our case study was between low (LSL = 5.0) and upper (USL = 5.08) specification limits. The process was deemed in control with the lower and upper control limits of LCL = 5.26 and UCL = 5.56. Then, based on existing records, causes of process failures and defects in construction plaster, which were connected with the plaster's qualitative characteristics, were examined using a "cause and effect" diagram" [41]. Finally, parameters that could improve customer satisfaction after identifying and prioritizing the foreseeable failure modes were determined and analyzed applying failure mode and effects analysis (FMEA) [42]. The statistical population of this research comprised "PoP", which is baked at a particular time in the "low burn" kiln and moved from baking salon to storage silo. The sampling method was a stratified random sampling method. Because of the characteristics of the plaster production process and consistent with the background studies, 25 subgroups of $n = 125$ samples were taken from multiple samples from different shifts. In this research, data were analyzed using three approaches of FMEA, ANN, and discriminant analysis (DA) [1]. To perform discriminant analysis, an understandable database for "SAS" software using Excel software was provided, and discriminant analysis was performed using programming ("Proc Discrim") in SAS software. For the case study of the present study, the data related to the critical parameter of the process were firstly collected, and the causes of product failure were identified and prioritized. Then, given that the proposed model is an intelligent hybrid model that can learn the patterns from input data (samples) using the power of learning neural networks, data were detected. Finally, the error of identifying training and test datasets was compared with the statistical method of discriminant

analysis. In this research, in order to monitor and troubleshoot the process, a model for combining SPC and artificial intelligence was designed using "MATLAB" software. The program codified in MATLAB is able to produce, present, and quickly encode neural network input data, as well as execute expert system rules. The program itself can also perform traditional SPC operations.

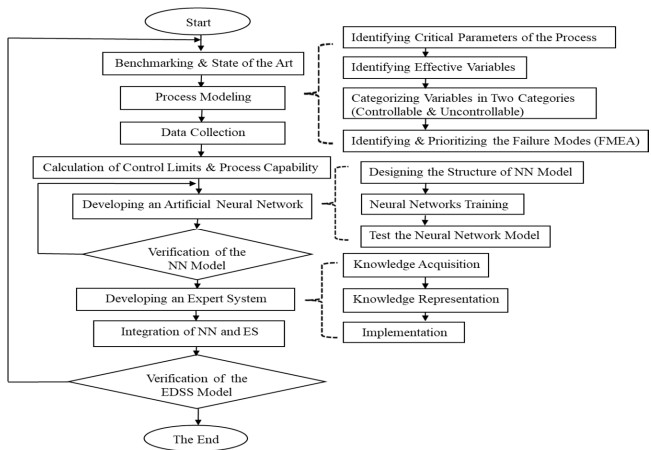

**Figure 4.** The flow diagram of the study procedure.

## 3. The Proposed Hybrid Fault Diagnosis Model

Based on what was said earlier, this research is based on the integration of NNs and ESs to provide analysis and interpretation for CCPs. The main focus of this study is to introduce a neural expert system-based pattern identifier, which will allow identifying abnormal patterns in order to correct their assignable causes. The operator will be warned if an abnormal pattern occurs in the process. By replacing human skills with a detection algorithm, human intervention is greatly reduced, and an intelligent manufacturing environment could be achieved. In this study, NNs are used to recognize control chart patterns, and an expert system is also used to interpret the identified pattern and determine the causes of the abnormal pattern. The general model of the research is depicted in Figure 5. As Figure 6 indicates, the proposed system consists of three subsystems:

- The SPC subsystem controls the traditional statistical process and, using statistical formulas, draws mean and variance for the sampled data of the process. It also sets control limits and determines the capability of the process and, in cases where any point on the charts is out of control, alerts "out of control" mode.
- The pattern recognition subsystem is accountable for detecting abnormal CCPs. Here, unnatural patterns in the X-bar chart are detected using neural networks, and abnormal patterns in the R chart are identified using "*Western Electric*" with the rule-based expert system.
- The reasoning subsystem is responsible for interpreting the purpose of process variations and proposing corrective or preventive actions. In this subsystem, using process-specific knowledge provided as if–then rules in the knowledge base, the cause of the abnormal patterns in the X-bar chart is interpreted. On the other hand, the cause of the unnatural patterns in the R chart is interpreted using general process knowledge, presented as if–then rules in the knowledge base (Figure 6).

Overall, the model design structure can be divided into three stages of neural network creation, expert system development, and integration of neural network and expert system, as explained below.

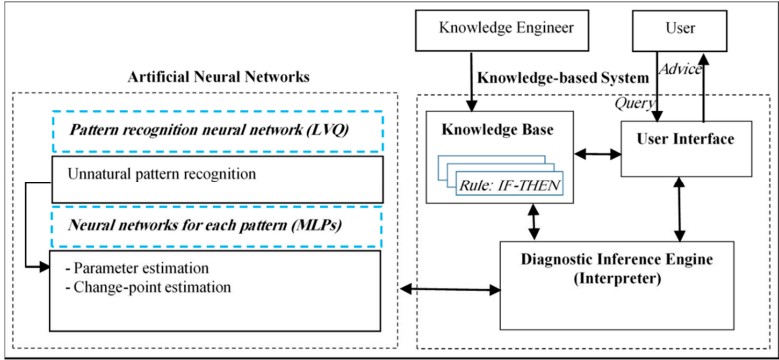

**Figure 5.** The structure of the neural expert system.

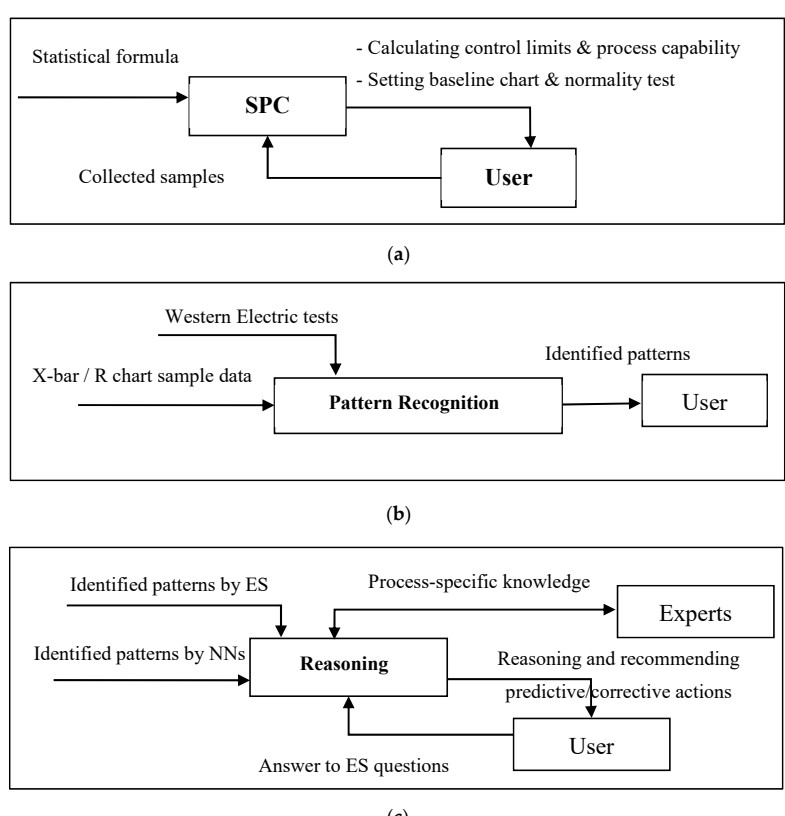

**Figure 6.** (**a**) SPC subsystem; (**b**) pattern recognition subsystem; (**c**) reasoning subsystem.

## 4. Experimental Results

This section describes the detailed characteristics of the structure of the proposed model and provides the results of their performances.

### 4.1. Developing the Neural Network Model

In the subsections below, the procedure for simulating normal and abnormal patterns in this research is firstly described. Then, the steps of creating a neural network, including neural network model structure design, neural network training, and neural network model validation, are presented.

### 4.1.1. Simulation of Unnatural CCPs and Their Corresponding Parameters

In the present research, because of the lack of a large number of useful samples to investigate abnormal CCPs, simulation of the models for training and testing networks was required; however,

it was attempted to simulate the data in line with the underlying process data. In the statistical issues, there is a probability distribution function for every random variable based on which the relevant parameters are also determined. Therefore, any natural deviations could be determined according to the probability distribution function of the corresponding random variable [12]. With these explanations, the parameters and functions employed to simulate control chart patterns are presented in Table 3. In this table, the parameters, on the one hand, represent the number of non-random disturbances. Furthermore, they reflect the process improvement during the implementation of recovery programs. In designing the proposed model, it is intended to identify the X-bar chart patterns using NN. The simulator function for the natural variation of the X-bar chart includes normal distribution: $x(t) = n(t)$, and the parameters of this distribution in our case are $\mu = 5.4$ and $\sigma = 0.1$. Given these values, the corresponding parameters of the other abnormal patterns in this chart were calculated, as shown in Table 3.

**Table 3.** Simulator functions of CCPs and the range of corresponding parameters' changes.

| Pattern Type | Simulator Functions | Parameter Change Range |
|---|---|---|
| Natural | $x(t) = n(t)$ | - |
| Shift (Sh.) | $x(t) = n(t) + u \times b$ [1] | $b = [1\sigma\sim3\sigma] \Rightarrow [0.1, 0.3]$<br>$b = [-3\sigma\sim-1\sigma] \Rightarrow [-0.3, -0.1]$ |
| Trend (Tr.) | $x(t) = n(t) + s \times t$ [2] | $s = [0.1\sigma\sim0.3\sigma] \Rightarrow [0.01, 0.03]$<br>$s = [-0.3\sigma\sim-0.1\sigma] \Rightarrow [-0.03, -0.01]$ |
| Cycles (Cyc.) | $x(t) = n(t) + l$ [3] $\times \sin((2\pi t)/T$ [4]$)$ | $l = [1\sigma\sim3\sigma] \Rightarrow [0.1, 0.3]$<br>$T = 8, 12, \ldots$ |
| Systematic (Sys.) | $x(t) = n(t) + g$ [5] $\times \cos(\pi t)$ | $g = [1\sigma\sim3\sigma] \Rightarrow [0.1, 0.3]$ |
| Shift + Trend (Sh. + Tr.) | $x(t) = n(t) + u \times b + s \times t$ | $b = [1\sigma\sim3\sigma] \Rightarrow [0.1, 0.3]$<br>$s = [0.1\sigma\sim0.3\sigma] \Rightarrow [0.01, 0.03]$<br>$b = [-3\sigma\sim-1\sigma] \Rightarrow [-0.3, -0.1]$<br>$s = [-0.3\sigma\sim-0.1\sigma] \Rightarrow [-0.03, -0.01]$ |
| Shift + Cycle (Sh. + Cyc.) | $x(t) = n(t) + u \times b + l \times \sin((2\pi t)/T)$ | $b = [1\sigma\sim3\sigma] \Rightarrow [0.1, 0.3]$<br>$l = [1\sigma\sim3\sigma] \Rightarrow [0.1, 0.3]$<br>$T = 8, 12 \ldots$ |
| Trend + Cycle (Tr. + Cyc.) | $x(t) = n(t) + s \times t + l \times \sin((2\pi t)/T)$ | $s = [0.1\sigma\sim0.3\sigma] \Rightarrow [0.01, 0.03]$<br>$l = [1\sigma\sim3\sigma] \Rightarrow [0.1, 0.3]$<br>$T = 8, 12 \ldots$ |

[1] Shift magnitude. [2] Trend slope (s). [3] Amplitude. [4] Period (T). [5] Magnitude of variations (g).

### 4.1.2. Designing the Structure of the Neural Network Model

In designing the general structure of the NN model, a modular approach was used. The overall model structure consisted of two separate sets of Module I and Module II. In the modular approach, the inputs and outputs of each network can be better managed, and the results of each network performance can be traced.

● **Module I**

Module I was developed to diagnose the behavior of the plaster production process. To this end, the classification power of competing algorithms was used, and a learning vector quantization (LVQ) network was designed to classify input patterns.

o Topology of LVQ Network

In the LVQ network (Figure 7), the connection type between layers is semi-connected, and the input vector, according to the process requirements, includes 25 neurons (25 samples taken from the process). The first layer contains 175 neurons, and the second layer includes eight neurons, while there was no need to consider the term bias. Each of the second-layer neurons represents one of the simulated patterns. Accordingly, neurons # 1 of the natural patterns and other neurons detect abnormal patterns of shift, trend, cycle, systematic, shift + trend, shift + cycle, and trend + cycle, respectively. Due to minimizing the number of outputs, there are patterns in this network, such as "upward shift" and

"downward shift", which represent a pattern with the same equations but values of different parameters. The main criterion for determining the number of subgroups required per class was reducing the incorrect identification of patterns. On the other hand, we attempted to assign almost identical neurons to patterns with the same number of parameters (Table 4).

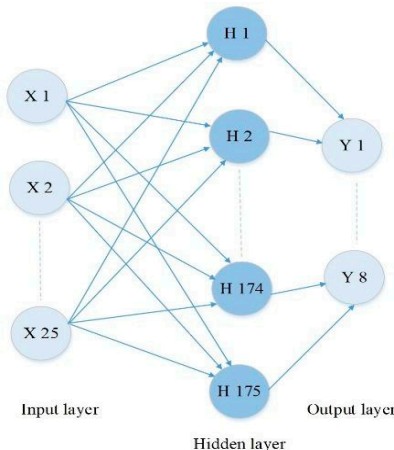

**Figure 7.** Learning vector quantization (LVQ) network.

**Table 4.** Number of inputs, hidden, and output layer neurons.

|  |  | Neuron |
| --- | --- | --- |
| **Input** |  | 25 |
|  | Natural (Network 1) | 1 |
|  | Upward Shift (Network 2) | 12 |
|  | Downward Shift (Network 2) | 12 |
|  | Upward Trend (Network 3) | 18 |
|  | Downward Trend (Network 3) | 18 |
| **Hidden Layer** | Cycles (Network 4) | 20 |
|  | Systematic (Network 5) | 4 |
|  | Upward Shift + Upward Trend (Network 6) | 18 |
|  | Downward Shift + Downward Trend (Network 6) | 18 |
|  | Shift + Cycles (Network 7) | 27 |
|  | Trend + Cycles (Network 8) | 27 |
|  | **Total** | 175 |
| **Output** |  | 8 |

o Learning Algorithm Used in Module I

The LVQ network in module I was trained by "enabling competition to take a place among the "*Kohonen*" neurons. The competition is based on the Euclidean distances ($d_i$) between the weight vectors ($W_i$) of these neurons ($i$) and the input vector ($x$).

$$d_i = \|Wi - X\| = \sqrt{\left(W_{ij} - X_j\right)^2}. \tag{1}$$

The neuron which has the least distance is the winner in the competition and is allowed to change its connection weights. The weights of the other neurons remain unchanged. The new weights can be obtained from

$$W_{\text{new}} = W_{\text{old}} + \lambda(X - W_{\text{old}}), \tag{2}$$

if the winner neuron is in the correct output category, or

$$W_{\text{new}} = W_{\text{old}} - \lambda(X - W_{\text{old}}), \tag{3}$$

if the winner is in the wrong category [18,27].

In the above equations, $\lambda$ is the learning rate, which decreases monotonically with the number of iterations. In this research, $\lambda = 0.01$ was considered. Appendix A provides the Matlab code.

- **Module II**

Module II in the proposed model was formulated to estimate the parameters of unnatural patterns of process control diagrams and estimation of the change point of the unnatural patterns.

o Topology of MLP Networks

In Module II, seven multilayer perceptron networks or MLPs do basic (single) and concurrent (mixture) pattern analysis. In this module, each of the networks in this set performs only the interpretation of one of the abnormal patterns. In these networks, the main parameters are estimated based on the definitions set out in Table 3. Moreover, in Module II, conditions for estimating the change point of abnormal behaviors in control charts are provided. In MLP networks (Figure 8), the type of layer connection is fully connected, and the number of inputs to all MLP networks is 26, 25 of which are the number of input neurons and one of which is the network bias which is equivalent to 1 for all simulated data. The number of hidden-layer neurons is optimized such that, for a certain amount of error for MLP on all network networks, the number of iterations required to achieve the error desired was calculated. Then, the number of neurons with the least number of repetitions until the desired error is chosen as the number of optimal neurons (Table 5). In each output layer, the neuron is embedded considering the number of related parameters of every pattern. In order to approximate different orientations of process changes, for example (upward or downward) given the outputs in the interval [−1, 1], a bipolar sigmoid function with the A = 0.1 constant is used. The relationship of the desired transfer function can be seen below.

$$g(x) = \frac{1 - e^{-x}}{1 + e^{-x}} \quad x = \text{A.net.} \tag{4}$$

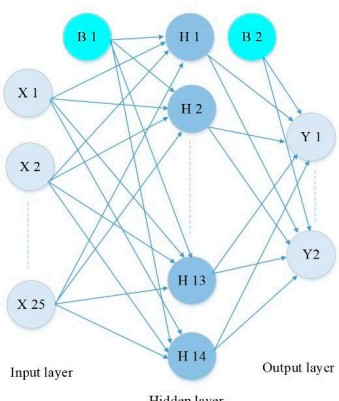

**Figure 8.** MLP network for shift pattern.

**Table 5.** Number of input, hidden, and output layer neurons.

| Network Name | Input | Hidden Layer | Output |
|:---:|:---:|:---:|:---:|
| Shift | | 14 | 2 |
| Trend | | 15 | 2 |
| Cycles | | 22 | 3 |
| Systematic | 25 + 1 (bias) = 26 | 17 | 2 |
| Shift + Trend | | 25 | 3 |
| Shift + Cycle | | 23 | 4 |
| Trend + Cycle | | 21 | 4 |

o Learning Algorithm Used in Module II

The training method in MLP networks is "backpropagation with an adaptive learning rate, where the weight of each layer, by the output and output derivative, is corrected until the network is fully trained". In this study, the training dataset is applied to corresponding networks in a category form, and errors are calculated at each step until the learning process is performed. Learning rate ($\lambda$) also changes according to the following command, so that the $E(t)$ is the network error at the time step $t$:

$$\lambda = \{0.99\lambda(t-1) \quad if \quad E(t) \leq E(t-1) 0.01\lambda(t-1) \quad if \quad E(t) > E(t-1). \tag{5}$$

There is a condition of training stop on the network error, which tries to minimize the error square between network outputs and the objective function using the gradient descent method. The network error, which is a cumulative error, is defined below in which $p$ stands for pattern number, $o$ represents the output neurons, $d_{ij}$ is desired value for the $j$ output of $i$ pattern, and $o\_ij$ is the actual output of the network for the $i$ pattern [43]. Appendix B provides the source code written in Matlab.

$$E = \frac{1}{2} \sum_{i=1}^{p} \sum_{j=1}^{o} \left(d_{ij} - o_{ij}\right)^2. \tag{6}$$

o Change Point of Unnatural Patterns

Estimation of the change point (starting point) and subsequent length of the unnatural pattern sequence expressed when the problem started and how long it lasted can help discover the causes of disorders. Since the neural network gives the change point of the abnormal patterns as a Module II output (MLP network) when generating training data, the change point is randomly generated between one and 10, and it is trained on the network to estimate its value as the network output. It should be noted that, in Module I, a fixed number is assumed to be the starting point for each pattern. In opting for the assumed fixed number as the change point, the conditions for pattern formation are considered with regard to its parameters. For example, since the cycle pattern has a period parameter (T = 8–12), the starting point should be earlier.

### 4.1.3. Neural Network Training

Training examples were introduced randomly to the NN. Before training, connection weights were generated with small random values. Weights were adjusted by training and presenting each pattern to the network. A maximum of 200,000 repetitions was considered as the stopping criterion for training. During the training phase, a series of vectors are provided to the network. The training vector consists of two sub-vectors: An input pattern and a target pattern. There are a total of 33 values for each training vector. A series of 25 coded observations, called the input pattern, is presented to the input layer, and the target pattern, which has an integer output of its inputs, is presented to the output layer (Table 6). Since each pattern has two orientations of changes (e.g., positive and negative shift), the desired output is set to 1 or −1. An output of 1 corresponds to a positive change, and an output of −1 corresponds to a negative change. For example, the output vectors of the "natural" pattern would be [10000000] and downward shift [0–1000000].

● **Training dataset**

In this study, the simulated data for the neural network model were divided into two subsets of training data and test data. Since there was no prior knowledge of the relative importance of unnatural patterns here, the training set contained approximately an equal number of training data for each type of pattern. In total, there were 11,000 training samples in the study set. The total number of training data in the LVQ network was 4000 samples, which equally considered 500 for each pattern. The total number of training samples for MLP networks was 7000, in which the same amount of 1000 samples was generated and applied for each of the seven MLP networks. In order to produce

the training dataset with the specifications mentioned above, a program was codified, which was capable of producing an unlimited number of natural and unnatural patterns with different parameters (for example, with different mean and standard deviations).

**Table 6.** Scaling range for outputs of MLPs.

|  | Pattern Type | Shift | Trend | Cycle | Systematic | Shift-Trend | Shift-Cycle | Trend-Cycle |
|---|---|---|---|---|---|---|---|---|
| Output 1 | min | −0.4 | −0.04 | 0.01 | 0.01 | −0.4 | 0.01 | 0.001 |
|  | max | 0.4 | 0.04 | 0.4 | 0.4 | 0.4 | 0.4 | 0.04 |
| Output 2 | min | 0 | 0 | 6 | 0 | −0.04 | 0.01 | 0.01 |
|  | max | 12 | 12 | 14 | 12 | 0.04 | 0.4 | 0.4 |
| Output 3 | min | - | - | 0 | - | 0 | 6 | 6 |
|  | max | - | - | 12 | - | 12 | 14 | 14 |
| Output 4 | min | - | - | - | - | - | 0 | 0 |
|  | max | - | - | - | - | - | 12 | 12 |

o Dataset scaling

In order to scale the dataset, the upper and lower boundaries of the input data were firstly specified by matching the maximum and minimum values of the input parameters. Then, the desired data were scaled and expanded into fitting values according to the type of the used transfer functions. Using the scaling method, all of the above operations were performed with the program written in Matlab. In the given formula, *A* is the "original value", *Ascale* is the "normalized value", *Amin* is the "minimum observable value", and *Amax* is the "maximum observable value". *Amin* and *Amax* might be estimated depending on the nature of data.

$$Ascale = min + \frac{max - min}{Amax - Amin}(A - Amin). \tag{7}$$

In the current application, the intervals used to scale the inputs of MLPs, considering the maximum and minimum ranges for the parameters of the unnatural patterns ($\pm\sigma3$) and given the mean ($\mu = 5.4$) and standard deviation ($\sigma = 0.1$), were [5.1, 5.7], which were scaled with confidence intervals of [3.4, 7.4]. In the LVQ network, the data scaling range (training and testing) was [−5, +5]. In this study, all input and output data of MLP networks were scaled in a [−1, +1] interval; however, before the output values were scaled in the [−1, +1] interval, each parameter was scaled to the separate maximum and minimum values. The different scaling values corresponding to the outputs of the MLPs are visible in Table 6. The maximum cumulative error (MCE) for training the LVQ was 0.047 (188 in 4000 training data), and that for testing was 0.0525. Table 7 shows the MCE and training iteration of each MLP network.

**Table 7.** Module I, evaluation results.

| Pattern Type | Direct Identification | Identification (Incomplete/Indirect) | Wrong Identification | Type I Error | Type II Error |
|---|---|---|---|---|---|
| Shift | 41 | 6/50 = 0.12 | 3/50 = 0.06 | 0.00 | - |
| Trend | 48 | 2/50 = 0.04 | 0.00 | 0.00 | 1/50 = 0.02 |
| Cycle | 48 | 1/50 = 0.02 | 1/50 = 0.02 | 0.00 | 1/50 = 0.02 |
| Systematic | 50 | 0.00 | 0.00 | 0.00 | 0.00 |
| Shift + Trend | 50 | 0.00 | 0.00 | 0.00 | 0.00 |
| Shift + Cycle | 50 | 0.00 | 0.00 | 0.00 | 0.00 |
| Trend + Cycle | 44 | 0.00 | 6/50 = 0.06 | 0.00 | 0.00 |
| **Total Error** | 21/400 = 0.052 | 9/400 = 0.022 | 10/400 = 0.025 | 0.00 | 2/400 = 0.005 |

### 4.1.4. Neural Network Test

After training the network with the training dataset, the network was evaluated by the test dataset. In the training phase, network efficiency was increased by minimizing errors between actual outputs.

In the test phase, solely the input vector was given to the network, where, in this case, the network was validated by predicting the response values for input and output.

● **Module I, Evaluation of LVQ Network**

For each input vector, the LVQ network decides about the production process situation. Therefore, a chance of errors in decision-making issues will arise. In case the network incorrectly recognizes the natural variation of the process abnormally, it commits a type I error. If it does not recognize the abnormal pattern in the process, a type II error takes place. An incorrect identification error occurs when random deviations cause the basic patterns in the early parts of the formation to have similar behavioral characteristics. The same will apply to concurrent patterns. As each random pattern warns of a particular disturbance in the process, incorrect pattern recognition has different costs. Moreover, if a basic abnormal pattern is identified in the form of a concurrent pattern comprising a basic pattern, it is considered indirect identification. On the other hand, if only one of the unnatural patterns is identified during the simultaneous occurrence of two abnormal patterns, the identification is putatively incomplete. The performance of Module I was measured according to the instructions and definitions performed by 400 test vectors, where each of them represents 25 samples of the plaster production process, which represents one of the eight patterns identified by the neural network. We applied each of the samples as input to the network and then compared the network response with the target response and calculated the network error rate. Table 7 presents the merged results for the 400 test vectors. As can be seen in the table below, the maximum LVQ network error in pattern recognition was 0.052 (21 in 400 data), which demonstrates that the proposed model was successful and effective due to the variety of trained patterns in the identification problem.

● **Module II, Evaluation the MLP Networks**

One of the important issues in neural network training is the overfitting problem of the training data. To put it bluntly, the network learns data very well, and it even remembers the noise in the data (disturbances)—excessive compliance—but it has serious problems identifying and generalizing new data [26]. To solve the problem, when the test data error increases while maintaining or decreasing errors related to training data, the training is stopped, and the final parameters are considered with the minimum error of the test data. The performance of the MLP networks in Module II was examined with numerous examples, and the results were satisfactory. As seen in Table 8, the calculated cumulative error of each MLP network was less than 0.02, which indicates that Module II was successful in identifying the parameters.

**Table 8.** Module II, evaluation results.

| Network Name | Maximum Cumulative Error (MCE) | Training Iteration | Minimum Number in Each Training | Hidden Layer Neurons | Output Neurons | Error of MLPs |
|---|---|---|---|---|---|---|
| Shift | 10 | 10 | 10,537 | 14 | 2 | 0.01 |
| Trend | 18 | 13 | 16,423 | 15 | 2 | 0.018 |
| Cycle | 25 | 10 | 35,360 | 22 | 2 | 0.016 |
| Systematic | 12 | 10 | 48,343 | 17 | 2 | 0.012 |
| Shift + Trend | 18 | 10 | 48,343 | 25 | 3 | 0.012 |
| Shift + Cycle | 25 | 11 | 46,703 | 23 | 4 | 0.012 |
| Trend + Cycle | 28 | 10 | 37,298 | 21 | 4 | 0.014 |

*4.2. Expert Systems*

In designing the general framework for the proposed expert system, a rule-based approach was used. ES assists quality control engineers, and it can be used for training operators as well. Therefore, the proposed system runs in three modes: A tutorial mode that offers explanation and training if requested by the user, a status mode that concludes from the evidence and responses provided

by the user, and a diagnosis mode that provides inference or reasoning with the rules within the knowledge base.

### 4.2.1. Knowledge Acquisition

"The knowledge acquisition process includes extracting, transforming, and validating expertise from different information sources for developing a knowledge base repository" [23]. The knowledge used in this research consists of "general knowledge" and "process-specific knowledge". In this study, to assist the fault diagnosis process, "Western Electric" [44] tests were utilized as general knowledge, and, to gather process-specific knowledge, "cause and effect (Ishikawa diagram)" [41] diagrams were prepared to investigate the root-cause. Using cause and effect diagrams, the most problematic reasons in the plaster production process were systematically determined (Figure 9). Next, failure modes and effects analysis (FMEA) (Table 9) was applied as an analytical method that incorporates the technology and experts' knowledge in identifying and prioritizing foreseeable failure modes of the process in order to eliminate or reduce their occurrence [42]. Finally, FMEA analysis results were collected in the knowledge base with the aim of making a complete reference for future issues. As can be seen in Table 9, FMEA uses the risk priority number (RPN) to evaluate the risk level of the process. RPN is calculated by multiplying the scores of three risk factors named occurrence (O), which is the frequency of the failure, severity (S), which is related to the effect of disruption on the system, and detection (D), which refers to the probability of detecting the failure. FMEA uses five scales or scores (1–5) to measure these factors.

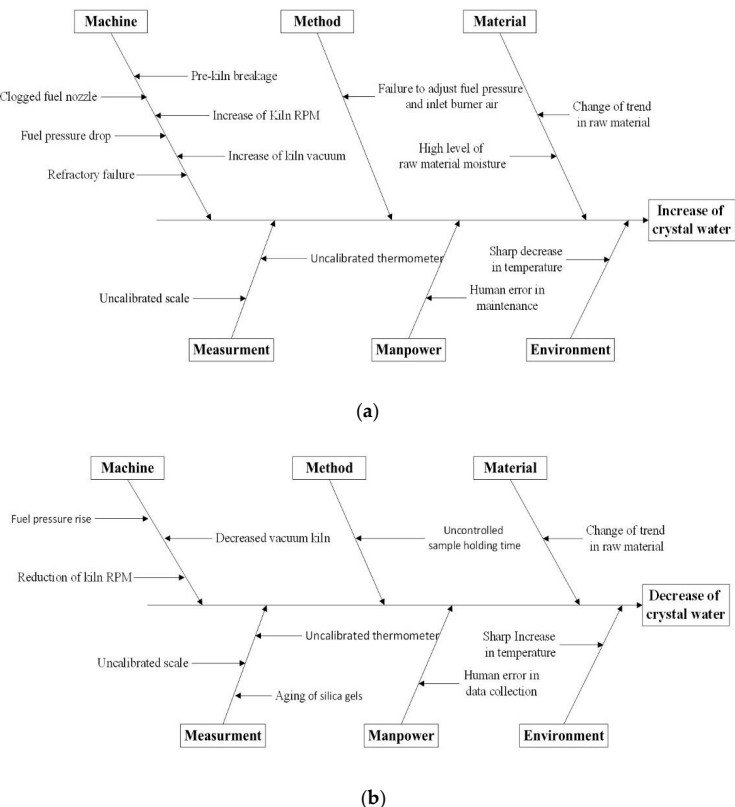

**Figure 9.** Cause and effect diagram: (**a**) for increase of crystal water; (**b**) for decrease of crystal water.

**Table 9.** Failure modes and effects analysis (FMEA) form for the critical parameter (crystal water).

| Failure Modes and Effects Analysis (FMEA) | | | | | | | |
|---|---|---|---|---|---|---|---|
| | Failure Mode | Index | Cause Effect | Corrective Actions | O | S | D | RPN [1] |
| 1 | | | Clogged fuel nozzle. | Cleaning of nozzle, establishment of PM for burner, and installation of filter. | 3 | 5 | 5 | 75 |
| 2 | | | Failure to adjust fuel pressure and inlet burner air. | Adjustment of fuel and air regulator in a defined period. | 3 | 4 | 4 | 48 |
| 3 | | | Pre-kiln breakage. | Chang or patching of pre-kiln and thickness monitoring. | 2 | 5 | 1 | 10 |
| 4 | | | Decrease of kiln temperature according to fuel pressure. | Installation of a shutdown sensor for fuel pressure. | 2 | 3 | 1 | 6 |
| 5 | Increase of Crystal Water | Decrease of middle temperature in the kiln body. | Increase of Kiln RPM. | Adjustment of RPM via frequency. | 1 | 5 | 4 | 20 |
| 6 | | | Sharp decrease in temperature. | - | 1 | 2 | 1 | 2 |
| 7 | | | Erosion in kiln blades | Change of runner blade and periodically monitoring. | 1 | 5 | 5 | 25 |
| 8 | | | Becoming dirty or clogging of the burner's air nozzle. | Cleaning the air nozzles and instating a multilayer filter. | 4 | 2 | 3 | 24 |
| 9 | | | Heat transition between kiln and environment because of the lake of refractory. | Establishment of PM and thickness monitoring of refractory. | 5 | 5 | 5 | 125 |
| 10 | | | Increase of negative pressure of kiln (filter). | Installation of ΔPMeter. | 5 | 5 | 1 | 25 |
| 11 | Decrease of Crystal Water | Increase of middle temperature in the kiln body. | Fuel pressure rise. | - | 1 | 2 | 1 | 2 |
| 12 | | | Decreasing of negative pressure of exhaust fan. | - | 5 | 5 | 3 | 75 |
| 13 | | | Reduction of kiln RPM. | - | 1 | 5 | 4 | 20 |
| 14 | | | Sharp Increase in temperature. | - | 1 | 2 | 1 | 2 |
| 15 | Variation in Crystal Water | - | Change of trend in raw material because of mine. | - | 4 | 5 | 4 | 80 |
| 16 | Increased Crystal Water | - | High level of raw material moisture. | - | 3 | 5 | 1 | 15 |
| 17 | Variation in Crystal Water | - | Changing of raw material spec. | - | 3 | 5 | 1 | 15 |

[1] RPN = O × S × D.

### 4.2.2. Knowledge Representation

In this study, a rule-based approach was considered to codify experts' problem-solving knowledge through inference rules: IF <a condition or premise>, THEN <an action or conclusion> rules. In total, 60 rules were used using technical documentation, operations procedures, and interviews with experts, for interpreting control charts (X-bar and R) and providing diagnostic expertise.

### 4.2.3. Implementation

In this study, the desired expert system was designed using three principal modules. The first module is related to knowledge base development, the second module is relevant to interface design and required questions to reach the answer, and the third module is associated with system run and dialogue to the user. Figure 10 shows a schematic of the proposed neural expert system and its components. Below is a brief description of each system component.

- A knowledge base uses the knowledge of experts and other sources acquired by knowledge engineers to support reliable, complete, and consistent decision-making in a time-critical situation. The knowledge base of the proposed model is an organized collection of facts and heuristics about the plaster production domain, as described briefly below.

    o "Facts" refers to a set of facts relating to the current process state extracted by a knowledge engineer (KE) from the records of the quality management system, preventive maintenance, calibration, brainstorming sessions, and interviews with experts.

o　　　"Procedures" focuses on manuals, standards, and procedures. Some examples include technical operation instructions, plaster production standards, and intelligent statistical process control (ISPC) tutorials.

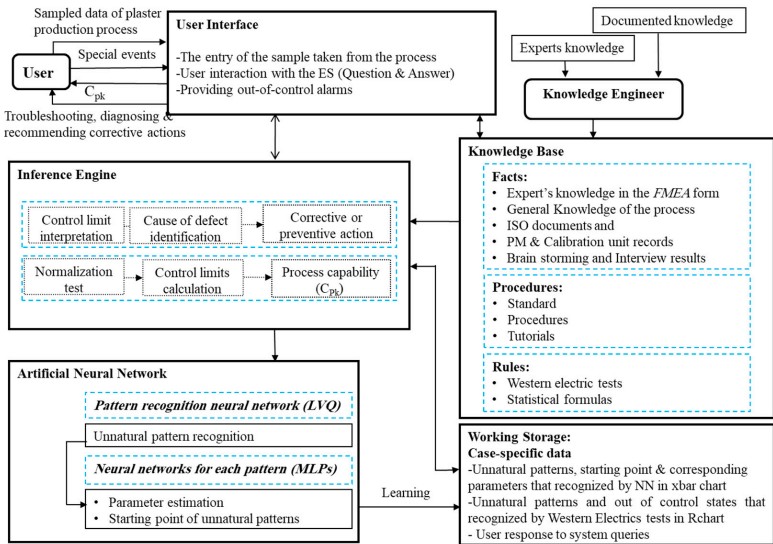

**Figure 10.** The proposed hybrid fault diagnosis system scheme or ISPC.

"Rules" relate to production rules that represent inferential knowledge learned from experts. In the knowledge base of the proposed model, 60 rules were employed that were extracted from interview with the experts, documents including "*Western Electric*" tests (general knowledge of the process), and the NN's response (case-specific data) extracted by KE and presented in the form if–then. Below is an example of a typical rule, based on specific knowledge of a process.

　　IF "diagnosis" is "upward trend",

　　AND "failure mode" is "increase of crystal water",

　　AND "process index" is "decrease of kiln's temperature",

　　THEN "specific cause" can be "clogged fuel nozzle",

　　AND "corrective actions" can be either "cleaning the fuel nozzle, the establishment of preventive maintenance (PM) for the burner, or installing fuel filter".

- The inference engine contains the inference strategies and matches the condition part of rules against facts of a specific case to reach a decision or conclusion. A backward chaining inference engine was used, as it is best suited for diagnosis-type systems, in which the codified program is executed with two groups of rules. The first group defines goals (assumed conclusions) for the properties and checks if their values are supported by the existing data. The second group updates the rules and transmits satisfying goals.
- The working memory acts as a repository for all data including the initial facts of the given case, the user's responses to system queries, and generated facts (e.g., type, change point, and parameters) derived by the inference engine.
- The user interface facilitates communication between the user and the proposed expert system through various input methods including dialog boxes and command prompts.

## 5. Comparative Analysis and Case Study

### 5.1. Comparison Study

To decisive the accuracy, consistency, and repeatability of test results, neural network model verification was done. The NN model was verified by comparing the error of the NN algorithm and

the error rate of the discriminant analysis (DA) method—a classification counterpart in the statistical approach [1]. The method of DA is to find a rule to separate two or more groups of observations from one another. The most important application of DA is classification. The output of the DA for the test dataset is presented in Table 10.

**Table 10.** Output of discriminant analysis (DA) for test dataset.

| **The SAS System (DISCRIM Procedure)** | | | | | | | | | |
|---|---|---|---|---|---|---|---|---|---|
| **Classification Summary for Test Data Set Using Linear Discriminant Function** | | | | | | | | | |
| **Number of Observations and the Percentage of Correctly Classified into the Target Classes** | | | | | | | | | |
| **Target** | **1** | **2** | **3** | **4** | **5** | **6** | **7** | **8** | **Total** |
| **1** | 17 | 15 | 8 | 0 | 0 | 8 | 0 | 0 | 48 |
| | 35.42 | 31.25 | 16.67 | 0.00 | 0.00 | 16.67 | 0.00 | 0.00 | 100.00 |
| **2** | 9 | 25 | 14 | 0 | 0 | 7 | 4 | 0 | 59 |
| | 15.25 | 42.37 | 23.73 | 0.00 | 0.00 | 11.86 | 6.78 | 0.00 | 100.00 |
| **3** | 10 | 12 | 23 | 1 | 0 | 6 | 0 | 0 | 52 |
| | 19.23 | 23.08 | 44.23 | 1.92 | 0.00 | 11.54 | 0.00 | 0.00 | 100.00 |
| **4** | 2 | 3 | 0 | 48 | 0 | 0 | 0 | 0 | 53 |
| | 3.77 | 5.66 | 0.00 | 90.57 | 0.00 | 0.00 | 0.00 | 0.00 | 100.00 |
| **5** | 0 | 0 | 0 | 0 | 50 | 0 | 0 | 0 | 50 |
| | 0.00 | 0.00 | 0.00 | 0.00 | 100.00 | 0.00 | 0.00 | 0.00 | 100.00 |
| **6** | 7 | 11 | 9 | 0 | 0 | 19 | 0 | 0 | 46 |
| | 15.22 | 23.91 | 19.57 | 0.00 | 0.00 | 41.30 | 0.00 | 0.00 | 100.00 |
| **7** | 0 | 0 | 0 | 0 | 0 | 0 | 39 | 0 | 39 |
| | 0.00 | 0.00 | 0.00 | 0.00 | 0.00 | 0.00 | 100.00 | 0.00 | 100.00 |
| **8** | 0 | 3 | 0 | 0 | 0 | 0 | 4 | 46 | 53 |
| | 0.00 | 5.66 | 0.00 | 0.00 | 0.00 | 0.00 | 7.55 | 86.79 | 100.00 |
| **Total** | 45 | 69 | 54 | 49 | 50 | 40 | 47 | 46 | 400 |
| | 11.25 | 17.25 | 13.50 | 12.25 | 12.50 | 10.00 | 11.75 | 11.50 | 100.00 |
| **Priors** | 0.12 | 0.1475 | 0.13 | 0.1325 | 0.125 | 0.115 | 0.0975 | 0.1325 | |

For example, in Table 10, the number 0 in the first row and the eighth column indicates that no "natural" pattern was mistakenly placed in the systematic patterns class. In the first row and the first column, the number 17 represents the number of patterns correctly assigned to the "natural" type. Furthermore, the value of 35.42 is the percentage correctly assigned to the "natural" class. In the first row and second column, 15 is the number of "natural" patterns that were mistakenly classified as "shift". The value of 31.25 is also the percentage of the "shift" pattern error in the "natural" class. The value of 31.25 is also the percentage of the "shift" pattern error in the "natural" class. Table 11 shows the errors for each class. This table lists the errors for each category in the "rate" line and the weight for each type in the "priors" row, while "total" (0.3325) indicates the total error of DA method for the test dataset. Tables 12 and 13 provide the output for the training dataset. As shown in the diagrams below (Figures 11 and 12), the NN outperformed the DA method in terms of performance and accuracy.

**Table 11.** DA error in the test dataset classification.

| **Error Estimation for Target Classes** | | | | | | | | | |
|---|---|---|---|---|---|---|---|---|---|
| | **1** | **2** | **3** | **4** | **5** | **6** | **7** | **8** | **Total** |
| **Rate** | 0.6458 | 0.5763 | 0.5577 | 0.0943 | 0.0000 | 0.5870 | 0.0000 | 0.1321 | 0.3325 |
| **Priors** | 0.1200 | 0.1475 | 0.1300 | 0.1325 | 0.1250 | 0.1150 | 0.0975 | 0.1325 | |

**Table 12.** Output of discriminant analysis (DA) for training dataset.

| Target | 1 | 2 | 3 | 4 | 5 | 6 | 7 | 8 | Total |
|---|---|---|---|---|---|---|---|---|---|
| **The SAS System (DISCRIM Procedure)** ||||||||||
| **Classification Summary for Training Data Set Using Linear Discriminant Function** ||||||||||
| **Number of Observations and the Percentage of Correctly Classified into the Target Classes** ||||||||||
| **1** | 218 | 175 | 124 | 1 | 0 | 0 | 0 | 0 | 518 |
|  | 42.08 | 33.78 | 23.94 | 0.19 | 0.00 | 0.00 | 0.00 | 0.00 | 100.00 |
| **2** | 162 | 227 | 116 | 3 | 0 | 21 | 8 | 2 | 539 |
|  | 30.06 | 42.12 | 21.52 | 0.56 | 0.00 | 3.90 | 1.48 | 0.37 | 100.00 |
| **3** | 158 | 158 | 163 | 1 | 0 | 21 | 0 | 0 | 501 |
|  | 31.54 | 31.54 | 32.53 | 0.20 | 0.00 | 4.19 | 0.00 | 0.00 | 100.00 |
| **4** | 5 | 24 | 6 | 471 | 0 | 0 | 11 | 1 | 518 |
|  | 0.97 | 4.63 | 1.16 | 90.93 | 0.00 | 0.00 | 2.12 | 0.19 | 100.00 |
| **5** | 4 | 1 | 0 | 0 | 489 | 0 | 0 | 0 | 499 |
|  | 0.80 | 0.20 | 0.00 | 0.00 | 98.00 | 0.00 | 0.00 | 0.00 | 100.00 |
| **6** | 49 | 25 | 42 | 0 | 0 | 314 | 0 | 0 | 430 |
|  | 11.40 | 5.81 | 9.77 | 0.00 | 0.00 | 73.02 | 0.00 | 0.00 | 100.00 |
| **7** | 0 | 1 | 0 | 13 | 0 | 1 | 399 | 1 | 415 |
|  | 0.00 | 0.24 | 0.00 | 3.13 | 0.00 | 0.24 | 69.14 | 0.24 | 100.00 |
| **8** | 0 | 0 | 0 | 5 | 0 | 56 | 42 | 477 | 580 |
|  | 0.00 | 0.00 | 0.00 | 0.86 | 0.00 | 9.66 | 7.24 | 82.24 | 100.00 |
| **Total** | 596 | 616 | 451 | 494 | 489 | 413 | 460 | 481 | 4000 |
|  | 14.90 | 15.40 | 11.28 | 12.35 | 12.23 | 10.33 | 11.50 | 12.03 | 100.00 |
| **Priors** | 0.1295 | 0.13475 | 0.12525 | 0.1295 | 0.12475 | 0.1075 | 0.10375 | 0.145 |  |

**Table 13.** DA error in the training dataset classification.

| | 1 | 2 | 3 | 4 | 5 | 6 | 7 | 8 | Total |
|---|---|---|---|---|---|---|---|---|---|
| **Error Estimation for Target Classes** ||||||||||
| **Rate** | 0.5792 | 0.5788 | 0.6747 | 0.0907 | 0.0200 | 0.2698 | 0.0386 | 0.1776 | 0.3105 |
| **Priors** | 0.1295 | 0.1348 | 0.1253 | 0.1295 | 0.1248 | 0.1075 | 0.1038 | 0.1450 |  |

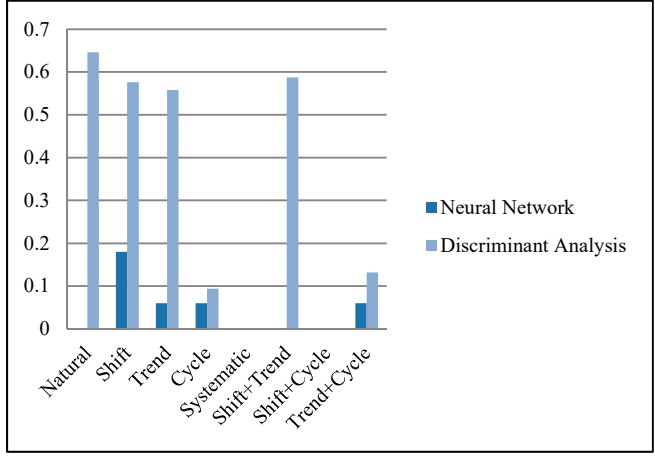

**Figure 11.** Comparison of DA and neural network (NN) error for each pattern in in test dataset.

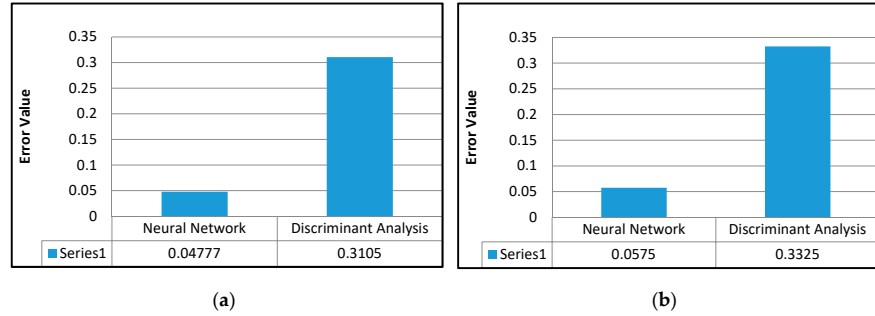

**Figure 12.** (**a**) Comparison of DA and NN error in test dataset; (**b**) comparison of DA and NN error in training dataset.

### 5.2. Case Study

In this section, to demonstrate the applicability and capability of the proposed model, a case study in a plaster-producing company is presented. In a traditional statistical process control system, after gathering the data, the following steps are done:

- Plotting the sequence of process measurements (observations).
- Setting UCL and LCL.
- Determining the process capability (Cpk).
- Performing the "normality test" on the data.
- Interpreting both R and X-bar chart for statistical control.

The proposed ISPC model, designed in Matlab, can adequately perform the above operations (Figure 13). Here, pursuant to the plaster-producing experts' opinion, if "Cpk > 1", the chart is considered as the baseline for interpreting the process. As can be seen in the ISPC implementation flow chart, after collecting the process data, the baseline chart should be set by eliminating and replacing points beyond the control limits of new data. As shown in Figure 14, the normality test was done, where the normality assumption was valid, and "Cpk > 1" in control mode. After drawing the baseline chart, the actual data of the process were inserted and checked and analyzed by the desired control charts. As illustrated in Figure 15, although there is no "out of control" mode within the "R-chart", the process was unable to meet specifications due to "Cpk < 1", being equal to 0.81 (Figure 16). On the other hand, by choosing the "X-bar chart" (Figure 17), the user receives the following error message: "X-bar chart is out of control" (Figure 18). Then, the ES using "Western Electric tests" announces that "out of control" modes may have the following reasons: "carelessness in the measurement, machinery stop, or off-spec materials". Later, the user receives a suggestion message from ES to check the unnatural patterns identified by NN (Figure 19). As can be seen in Figure 20, not only was the "downward shift pattern" in the "X-bar chart" identified by the NN, but the "starting point" of the unnatural pattern was estimated (point 6). The "shift magnitude parameter" (−0.161) was also determined. In this scenario, because of the appearance of a "downward shift pattern" and based on user observation, which was "kiln body scarlet", the reason for the deviation was recognized as the "temperature exchange of kiln with the environment due to the loss of refractory and thickness". "Establishment of maintenance and inspection of refractory" was also recommended as corrective or preventive activities. In this scenario, by making corrective actions and following re-sampling the process (Figure 21), "out of control" modes did not appear in the control charts anymore (Figure 22) and, furthermore, "process capability" increased from Cpk = 0.81 to Cpk = 1.15 (Figure 23). The experimental results show that corrective actions could significantly contribute to process recovery. Thus, the proposed fault diagnosis system could be used to support decision-makers of the plaster production.

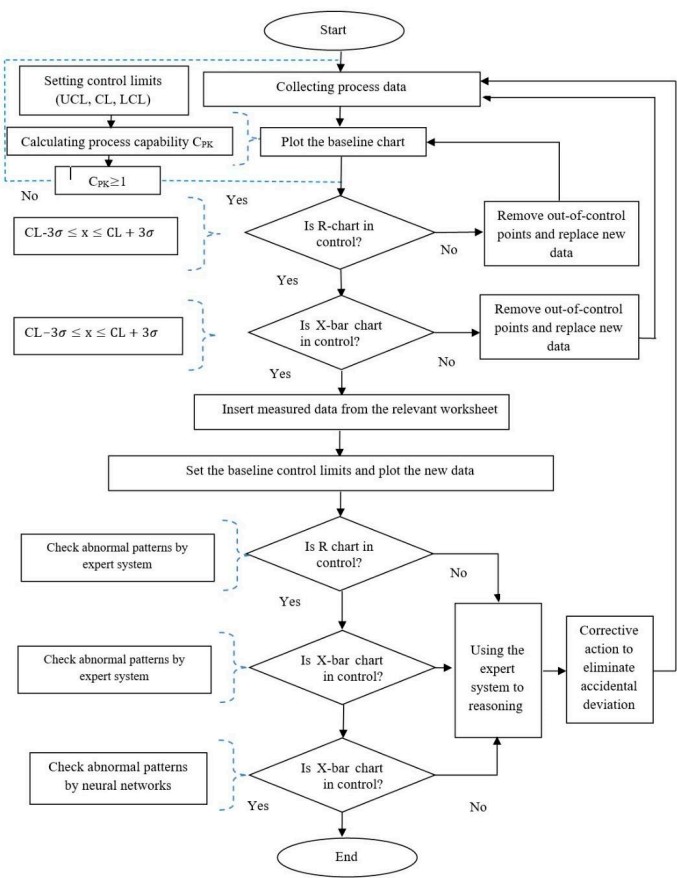

**Figure 13.** ISPC implementation flow chart.

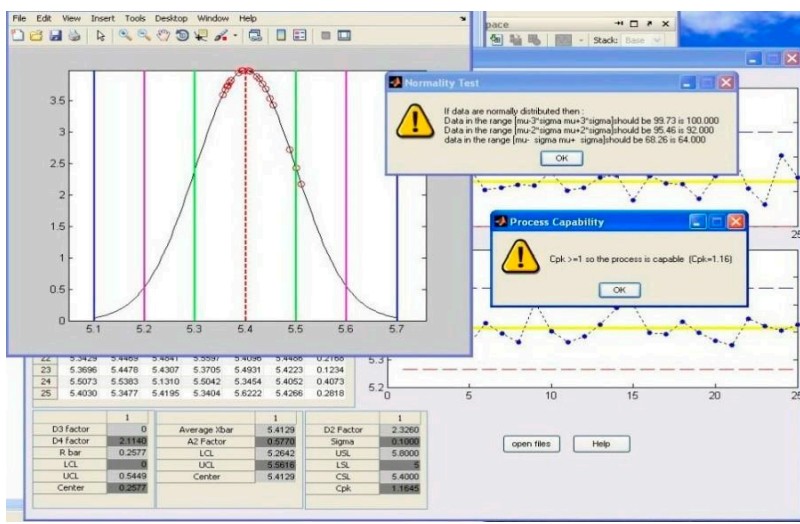

**Figure 14.** Normality test and process capability calculation.

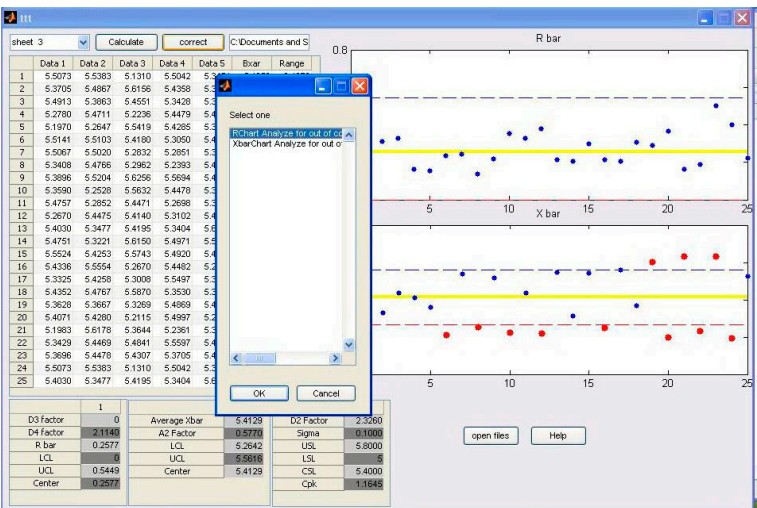

**Figure 15.** X-bar chart in "out of control" mode.

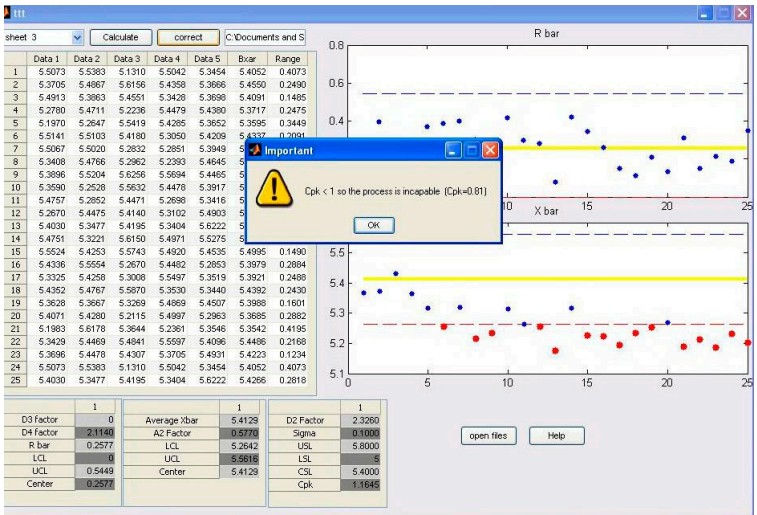

**Figure 16.** Process capability calculation.

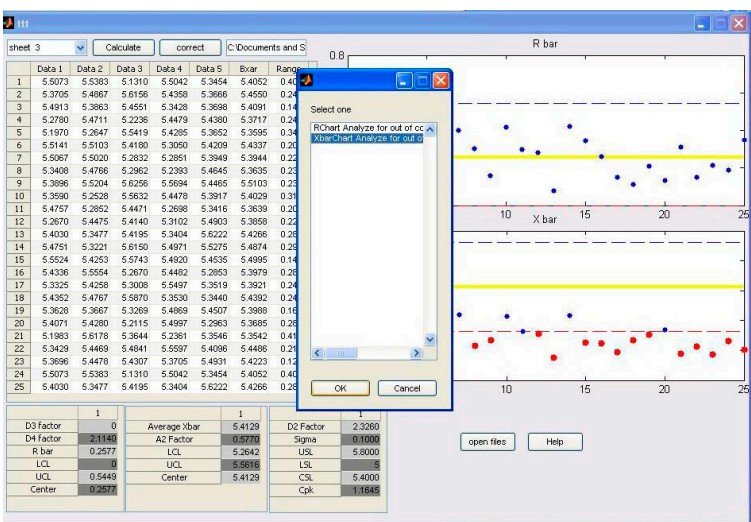

**Figure 17.** Request for analyzing the X-bar chart.

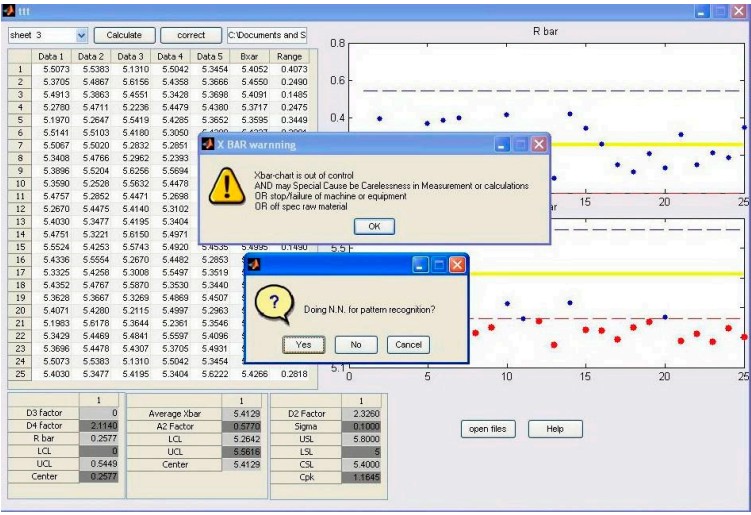

**Figure 18.** X-bar chart reasoning by Western Electric tests.

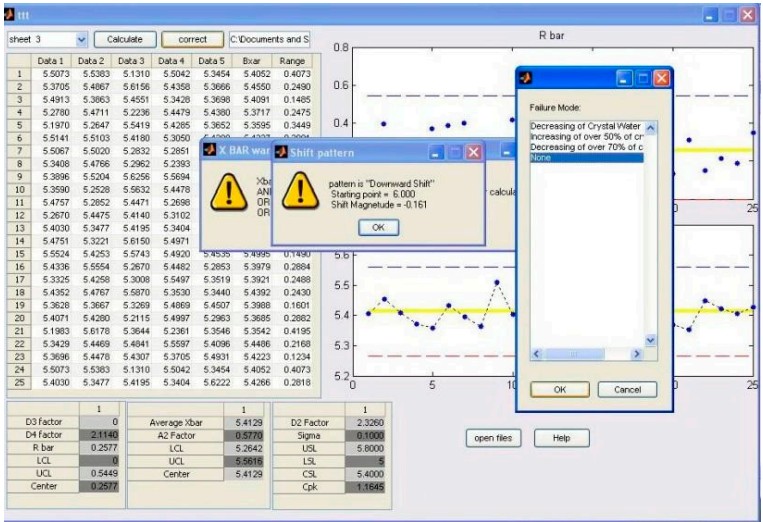

**Figure 19.** Pattern recognition in X-bar chart by NN.

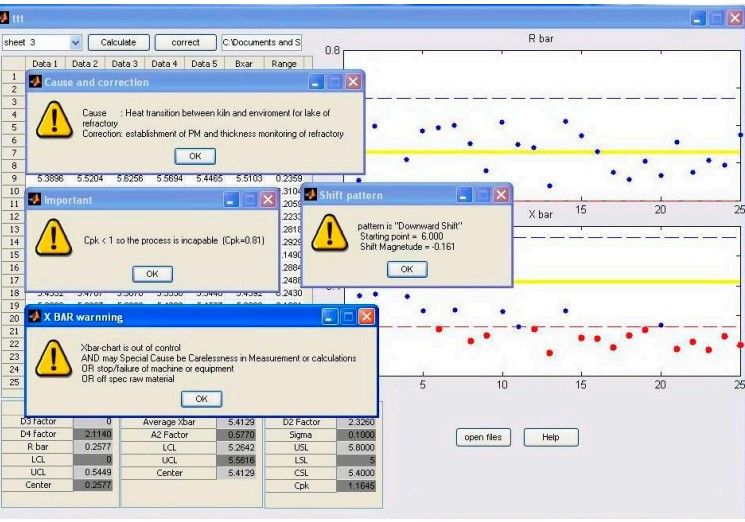

**Figure 20.** Determining Cpk, unnatural pattern, parameters, and recovery actions.

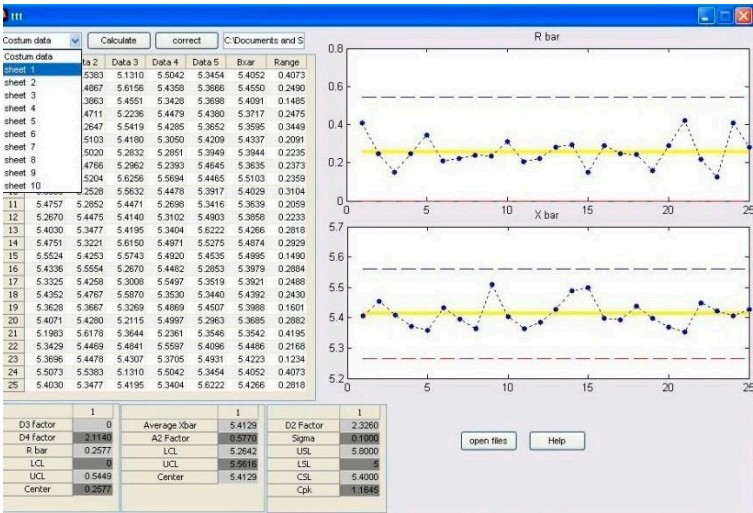

**Figure 21.** Inserting new dataset.

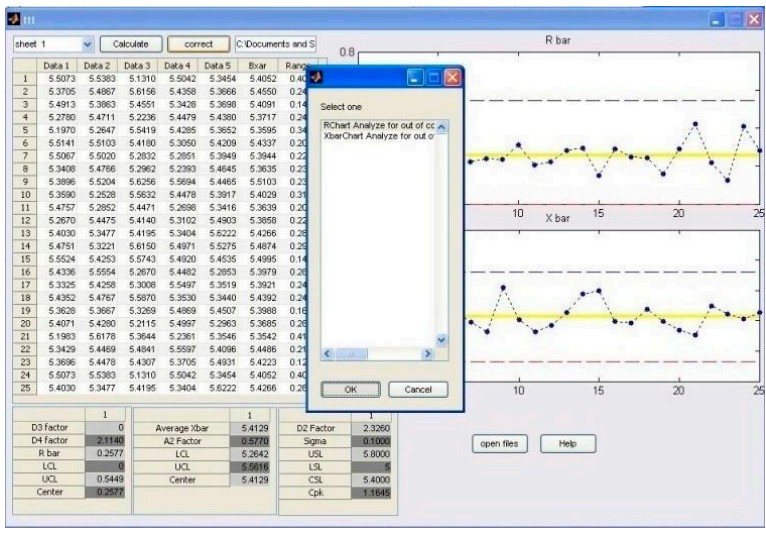

**Figure 22.** X-bar and R charts in control mode.

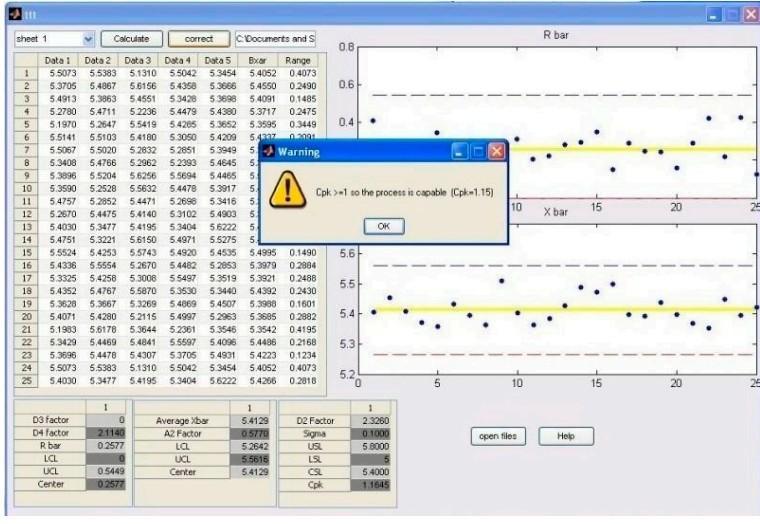

**Figure 23.** Capability of the process after performing corrective actions.

## 6. Conclusions

This paper aimed to target one of the most challenging subjects in smart manufacturing, which is the quality control at the shop floor level considering emerging technologies. There are many conceptual models and general recommendations when discussing a new paradigm of quality associated with I4.0, but there are relatively few works in action. The hybrid model proposed in this work supports the troubleshooting of the plaster production process, which is a complex manufacturing system. To have both descriptive and prescriptive approaches, NN and ES were integrated where NN deals with the determination of fault areas, and ES provides the recommendation of corrective actions.

The main achievements and contributions of this work are as follows:

1. Successful implementation of Quality 4.0 to blend traditional quality control models based on CCPs with an intelligent system at the shop floor level.
2. According to Tables 3 and 5, the diagnosis of behavioral patterns coming from Module I is acceptable, and parameters of corresponding patterns estimated by Module II are effective and reliable.
3. It was a multitask project including production, delivery, and encryption of neural network input.
4. Using a wide range of data in training the NNs to assure stable behavior in performance.
5. Using the integrated system, most SPC requirements, such as drawing of basis chart, checking X-bar and R charts for being under control, and calculating Cpks, were achieved.
6. Using LVQ for pattern classification and MLP in parallel. This helps in simultaneously enjoying the competitive power of the LVQ network and the interoperability of multilayer perceptron networks.
7. The result of the case study shows the improvement of process capability, while control charts did not show any out of control mode after following the corrective actions; thus, the capability of the proposed model to serve as a reliable decision support system (DSS) was confirmed.

This paper shows the capability of I4.0 to change the quality paradigm in factories of the future. The key element is the level of intelligence of the system, which leads to smart manufacturing. There is no doubt that emerging technologies will shift quality processes to a different level, while monitoring, fault detection, cause, root analysis, and even corrective actions and strategies would be autonomous.

For further research, there are many potential areas of working, as outlined below.

1. Developing and comparing the result of new models based on adaptive neuro-fuzzy inference systems (ANFIS).
2. Using the particle swarm optimization (PSO) to improve the performance.
3. Using some techniques and algorithms such as deep learning to increase the efficiency of the model.
4. Using collective sensor networks and Internet of things (IoT) platforms to develop a real-time smart quality control system.
5. Connecting smart quality to new services such as predictive maintenance to achieve smart, collaborative devices and support new product and production lines based on information from intelligent quality information.

The models proposed in this paper were independent of software; thus, free software and modern scripting languages such as Python, Ruby, etc. could be utilized for the same purpose.

This paper was a real case of Quality4.0 in action to show the capabilities and applicability of emerging technologies and intelligent algorithms to shift control quality to the new stage, and it represents the initial step of a long journey.

**Author Contributions:** Conceptualization, J.R.; methodology, J.R. and J.J.; writing—drafting Section 2, J.R. and J.J.; writing—drafting Section 3, J.R. and J.J.; writing—drafting Section 4, J.R.; writing—drafting Section 5, J.R.; writing—drafting Section 6, J.J.; writing—reviewing and editing, J.R. and J.J. All authors read and agreed to the published version of the manuscript.

**Funding:** This research was funded in part by the Portuguese "Fundação para a Ciência e a Tecnologia" (FCT) in the context of the Center of Technology and Systems CTS/UNINOVA/FCT/NOVA, reference UIDB/00066/2020.

**Acknowledgments:** This work was supported by the Portuguese Foundation for Science and Technology (FCT) and the Center of Technology and Systems (CTS).

**Conflicts of Interest:** The authors declare no conflict of interest.

## Appendix A

```
for l = 1:Ntraining;
for i = 1:Nhidden
for j = 1:Ninput
D(i,j) = (Data(l,j)-W(j,i))^2;
end
Distance(i,:) = [sqrt(sum(D(i,:),2)),i];
end
MinDis = min(Distance(:,1));
ONN = find(Distance(:,1) == MinDis);
for i = 1:Nhidden
if i == ONN
HiddenOut(i) = 1;
else
HiddenOut(i) = 0;
end
end
NetOut(l,:) = HiddenOut*V;
if NetOut(l,:) == RealOut(l,:)
W(:,ONN) = W(:,ONN) + Lambda*(transpose(Data(l,:))-W(:,ONN));
else
W(:,ONN) = W(:,ONN)-Lambda*(transpose(Data(l,:))-W(:,ONN));
end
Er(l) = sum(abs(RealOut(l,:)-NetOut(l,:)),2);
end
t = t + 1;
Error(t) = sum(Er,2)
if Error(t) < 500
Lambda = 0.99*Lambda;
End
```

## Appendix B

```
while Error(l) > Emax
l = l + 1;
Epoch(l) = l;
HO = V0*transpose(InData);
HiddenOut = (1-exp(-A*HO))./(1 + exp(-A*HO));
DiffHiddenOut = (A/2)*(1-HiddenOut.*HiddenOut
HiddenOut(Nshifthidden,:) = 1;
DiffHiddenOut(Nshifthidden,:) = (A/2)*(1-HiddenOut(Nshifthidden,:).*HiddenOut(Nshifthidden,:));
OO = W0*HiddenOut;
Output = (1-exp(-A*OO))./(1 + exp(-A*OO));
DiffOutput = (A/2)*(1-Output.*Output);
DeltaO = (transpose(OutData)-Output).*DiffOutput;
```

```
DeltaH = DiffHiddenOut.*transpose(transpose(DeltaO)*W0);
E = (OutData-transpose(Output)).*(OutData-transpose(Output));
for i = 1:Nshifthidden
for j = 1:Nshiftinput
V(i,j) = V0(i,j) + Lambda*DeltaH(i,:)*InData(:,j);
end
end
for i = 1:Nshiftoutput
for j = 1:Nshifthidden
W(i,j) = W0(i,j) + Lambda*DeltaO(i,:)*HiddenOut(j,:)';
end
end
V0 = V;W0 = W;
Error(l) = 0.5*sum(sum(E,1),2);
```

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
