# Peer review of "Quality 4.0 in Action: Smart Hybrid Fault Diagnosis System in Plaster Production"

_processes, doi:10.3390/pr8060634_

Round 1

Reviewer 1 Report

This paper proposed an smart hybrid fault diagnosis system based on ANN and ES. The paper is well writen and has significant instructions to industry application. Suggestions and comments for this paper are as follows:

1).It would be better to use subscript in Eq. 3 and Eq. 4. Using 'Wnew' instead of 'Wnew' etc. Please also check other equations.

2). Please add the explaination of the paramaters in the equations, like the 'Amax' in Eq. 7. Please also check other equations.

Author Response

This paper proposed an smart hybrid fault diagnosis system based on ANN and ES. The paper is well writen and has significant instructions to industry application. Suggestions and comments for this paper are as follows:

We very much appreciate the time and careful analysis made by the reviewer as well as the recommendations that helped us improve our paper.

1). It would be better to use subscript in Eq. 3 and Eq. 4. Using 'Wnew' instead of 'Wnew' etc. Please also check other equations.

It is adjusted according to the suggestion.

2). Please add the explaination of the paramaters in the equations, like the 'Amax' in Eq. 7. Please also check other equations.

It's explained in the new version.

Reviewer 2 Report

The paper under review considers the issue of quality 4.0 in action: Smart Hybrid Fault Diagnosis System in Plaster Production.

In the reviewer’s opinion the paper is very interesting.  However, there are several important aspects that require authors comments  or possibly improvements:

1) The authors of the article combine the "old SPC method" with the "modern methods" of nauron networks. Are there other methods worthy of attention between the SPC and NN methods? Can we combine other methods with SPC, e.g. other statistical methods instead of NN?

2) The authors should improve the quality of all figures as much as possible.

3) The authors developed the results based on commercial programs. Can we use other free software? or modern scripting languages such as python, ruby etc.

4)  Deep learning and deep networks are currently very popular.  Is the application of this approach worth attention from a practical point of view?

Author Response

The paper under review considers the issue of quality 4.0 in action: Smart Hybrid Fault Diagnosis System in Plaster Production.

In the reviewer’s opinion the paper is very interesting.  However, there are several important aspects that require authors comments or possibly improvements:

We very much appreciate the positive feedback and time and careful analysis made by the reviewer as well as the recommendations that helped us improve our paper.

1) The authors of the article combine the "old SPC method" with the "modern methods" of nauron networks. Are there other methods worthy of attention between the SPC and NN methods? Can we combine other methods with SPC, e.g. other statistical methods instead of NN?

In the paper already two methods, a statistical method called DA and NN were used, and the results are compared.

Section 5.1., Comparison Study
2) The authors should improve the quality of all figures as much as possible. We did our best improve the quality of the figures in the new version
3) The authors developed the results based on commercial programs. Can we use other free software? or modern scripting languages such as python, ruby etc.

The below explanation has added to the conclusion.

The models proposed in this paper were independent of software, so free software and modern scripting languages such as python, ruby, etc. could be utilized for the same purpose.
4)  Deep learning and deep networks are currently very popular.  Is the application of this approach worth attention from a practical point of view? The comment is not clear for the authors, but deep learning is suggested in conclusion and for future works/studies!

Reviewer 3 Report

The problem is well set and practically explained. However "Quality 4.0" is not a well-known term regarding Industry 4.0, thus in the article should be included more recent references. From 34 references only 6 are recent (published in last 3 years). Chapter 2 is quite a short and actually makes part of introduction.

The methodology chapter 3 referres ony to one reference [31], thus the headline "Materials and methods" is deceiving - there is single material as plaster involved and single method referred.  

The research part in Chapter 4 is named as "Proposed method", thus indicating the research done. The chapter is long and mixes original research and case study analysis. Regarding proposed method description in chapter 4 the Figures 14-23 are poorly readable and mostly illustrative screenshots not adding scientific value. These should be replaced by proper readable graphs. It would be recommended to split such practical example into separate chapter of analysis.

Conclusions part is well made and grounded.

Appendix A and B add practical value.

Regarding references - besides novelty there is also typho issue - reference 26 needs to be revised.

Author Response

The problem is well set and practically explained. We very much appreciate the time and careful analysis made by the reviewer as well as the recommendations that helped us improve our paper.

However "Quality 4.0" is not a well-known term regarding Industry 4.0, thus in the article should be included more recent references.

Already some more text and references added.

From 34 references only 6 are recent (published in last 3 years).

The references are updated:

 16 published in last 3 years
Chapter 2 is quite a short and actually makes part of introduction.

Following the suggestion of the reviewer, chapters 1 & 2 are merged.

The methodology chapter 3 referres ony to one reference [31], thus the headline "Materials and methods" is deceiving - there is single material as plaster involved and single method referred. 

The relevant references added to the paper.

The research part in Chapter 4 is named as "Proposed method", thus indicating the research done. The chapter is long and mixes original research and case study analysis. Regarding proposed method description in chapter 4 the Figures 14-23 are poorly readable and mostly illustrative screenshots not adding scientific value. These should be replaced by proper readable graphs. It would be recommended to split such practical example into separate chapter of analysis.

This section is divided into two different sections:

4. Experimental Results

5. Comparative Analysis and Case Study

Conclusions part is well made and grounded.

NA

Appendix A and B add practical value.

NA

Regarding references - besides novelty there is also typho issue - reference 26 needs to be revised.

Typho issue in reference 26 is revised.

Reviewer 4 Report

Thank you for giving me the possibility of reviewing this paper. I hope the authors find my comments productive and that it will help them to improve their research work.

In this article, the authors propose a Hybrid model based on Neural Network (NN) and Expert System (ES) dealing with control chart patterns (CCPs). Their objective is to have more than a descriptive passive model, smart predictive model to recommend corrective actions. A Construction Plaster producing company was used to present & evaluate the advantages of this novel approach while the result shows the competency & eligibility of Quality 4.0 in action

The objective is clearly stated.

The literature review is weak. More focus on the relevance of their research would be necessary to frame the innovativeness and relevance of their work. Figures and facts of the need of their model for industry 4.0. This has to be justified with adequate references to previous work.

The research experiment es clearly explained and well structured.

How have their results improved the previous body of knowledge? Relate the results with prior background.

For example, Rodríguez-Gonzalez, S., & Rivas, A. (2019). Neuro-Symbolic Hybrid Systems for Industry 4.0: A Systematic Mapping Study. In Knowledge Management in Organizations: 14th International Conference, KMO 2019, Zamora, Spain, July 15-18, 2019, Proceedings (Vol. 1027, p. 455). Springer.

The following paper might be interesting for the authors to read and refer so that they can find a way to justify implementation of their research for 4.0 businesses  

Palos-Sanchez, P., Reyes-Menendez, A., & Saura, J. R. (2019). Modelos de Adopción de Tecnologías de la Información y Cloud Computing en las Organizaciones. Información tecnológica30(3), 3-12.

The authors need to improve the conclusions making reference to possible implications of their work both related to researchers and practitioners.

Author Response

Thank you for giving me the possibility of reviewing this paper. I hope the authors find my comments productive and that it will help them to improve their research work. We very much appreciate the time and careful analysis made by the reviewer as well as the recommendations that helped us improve our paper.

In this article, the authors propose a Hybrid model based on Neural Network (NN) and Expert System (ES) dealing with control chart patterns (CCPs). Their objective is to have more than a descriptive passive model, smart predictive model to recommend corrective actions. A Construction Plaster producing company was used to present & evaluate the advantages of this novel approach while the result shows the competency & eligibility of Quality 4.0 in action

The objective is clearly stated.
NA
The literature review is weak. More focus on the relevance of their research would be necessary to frame the innovativeness and relevance of their work. Figures and facts of the need of their model for industry 4.0. This has to be justified with adequate references to previous work.  Some more text and relevant references are added to strengthen this part.
The research experiment es clearly explained and well structured. NA

How have their results improved the previous body of knowledge? Relate the results with prior background.

For example, Rodríguez-Gonzalez, S., & Rivas, A. (2019). Neuro-Symbolic Hybrid Systems for Industry 4.0: A Systematic Mapping Study. In Knowledge Management in Organizations: 14th International Conference, KMO 2019, Zamora, Spain, July 15-18, 2019, Proceedings (Vol. 1027, p. 455). Springer.
The contribution of the paper is described in the introduction.

The following paper might be interesting for the authors to read and refer so that they can find a way to justify implementation of their research for 4.0 businesses 

Palos-Sanchez, P., Reyes-Menendez, A., & Saura, J. R. (2019). Modelos de Adopción de Tecnologías de la Información y Cloud Computing en las Organizaciones. Información tecnológica, 30(3), 3-12.

It seems an interesting paper, but as it is in Spanish, it could not be used due to the lack of language knowledge of the authors.

The authors need to improve the conclusions making reference to possible implications of their work both related to researchers and practitioners.

Considering other reviewers' feedback & mainly the third reviewer, the conclusion had no major changes, but some minor changes due to the other comments are implemented.